# REVISITING GRADIENT EPISODIC MEMORY FOR CONTINUAL LEARNING

## ABSTRACT

Gradient Episodic Memory (GEM) is an effective model for continual learning, where each gradient update for the current task is formulated as a quadratic program problem with inequality constraints that alleviate catastrophic forgetting of previous tasks. However, practical use of GEM is impeded by several limitations: (1) the data examples stored in the episodic memory may not be representative of past tasks; (2) the inequality constraints appear to be rather restrictive for competing or conflicting tasks; (3) the inequality constraints can only avoid catastrophic forgetting but can not assure positive backward transfer. To address these issues, in this paper we aim at improving the original GEM model via three handy techniques without extra computational cost. Experiments on MNIST Permutations and incremental CIFAR100 datasets demonstrate that our techniques enhance the performance of GEM remarkably. On CIFAR100 the average accuracy is improved from 66.48% to 68.76%, along with the backward (knowledge) transfer growing from 1.38% to 4.03%.

## 1 INTRODUCTION

*Catastrophic forgetting* (McCloskey & Cohen, 1989) is a common phenomenon in deep learning that the model performance over past tasks can be harmed by the training process of a current task when we employ one single neural network to learn consecutive tasks. *Continual learning* (Ring, 1994), also known as *lifelong learning* (Thrun, 1994), is a specific research field in AI that focuses on avoiding or alleviating catastrophic forgetting. It is well known that humans and large primates can continually learn new skills and accumulate knowledge throughout their lifetime (Fagot & Cook, 2006).

Gradient Episodic Memory (GEM) (Lopez-Paz & Ranzato, 2017) is a significant method for continual learning. The basic idea is to modify each gradient of the current task such that the losses at the memories (data examples) from previously solved tasks will not increase, hence allowing positive backward knowledge transfer. The problem is formulated as a Quadratic Program with inequality constraints, and then solving its dual problem provides an efficient computation for a small number of dual variables. GEM outperforms two state-of-the-art methods, Elastic Weight Consolidation (EWC) (Kirkpatrick et al., 2017) and iCARL (Rebuffi et al., 2017) on variants of the MNIST and CIFAR100 datasets.

Despite the elegance of its mathematical formulation and the computational efficiency in solving the dual problem, the GEM model has following drawbacks for practical use.

1. The episodic memory is simply populated with the last $m = M/T$ examples from each task in their current implementation, where $M$ is the total budget size of memory locations for totally $T$ tasks. The authors of GEM mentioned that better memory update strategies could be employed to obtain a representative set of examples for past tasks, such as building a coreset per task (Lucic et al., 2018).

2. The inequality constraints of gradients appear to be quite restrictive, requiring that the gradient update can not increase any loss of all previously solved tasks. When two tasks are competing or conflicting in the gradient update directions, especially with a large number of past tasks, these constraints have to be violated and the feasible solution might be null.

3. The gradient constraints only assure to avoid catastrophic forgetting, but might not promote positive backward knowledge transfer in some scenarios. For instance, we may obtain a gradient vector orthogonal to all gradients of previous tasks in the constrained optimization problem. In this case, GEM is greedy in minimizing the loss of the current task, overlooking the need of backward transfer to previous tasks. An ideal model for continual learning should consider a better trade-off between optimizing performance of the current task and backward transfer to previously solved tasks.

To address these issues, in this paper we propose three techniques that improve the performance of GEM. These techniques come from successful practice in Support Vector Machines (SVMs) and deep learning approaches. In concrete, we use the notion of *margins* to select representative data points as episodic memories. We also leverage slack variables to allow violations of the constraints, and aggressively encourage backward knowledge transfer to previous tasks. Experiments on MNIST and CIFAR100 datasets corroborate the improvements of our techniques over the original GEM model. Also our methods increase little overhead in computational efforts.

The paper is organized as follows. In Section 2, we first survey several recent methods on continual learning and explain why we focus on improving GEM. Section 3 offers a brief review of the GEM algorithm. In Section 4, we elaborate our three improvements and discuss the underlying motivations. Section 5 presents the experimental results and comparison analysis, and Section 6 concludes the paper.

## 2   RELATED WORK

According to Xu & Zhu (2018), recent methods for continual learning can be roughly classified into two large categories, distinguished by whether the network architecture changes through a continuum of tasks.

The first category attempts to modify the network architecture to alleviate catastrophic forgetting. Fernando et al. (2017) proposed a neural network model called PathNet, in which there are ten or twenty modules in each layer and for one task only three or four modules are dynamically chosen in each layer using an evolutionary approach. Progressive neural networks (Rusu et al., 2016) aims at accommodate the new coming task by dynamically expand the architecture with a fixed size of nodes or layers, leading to an extremely large network structure with high redundancy. In Dynamically Expandable Network (DEN) (Yoon et al., 2017) this issue is slightly alleviated by employing group sparsity regularization when expanding the networks. However many hyperparameters are involved, including various regularization and thresholds, and careful tuning is challenging. Xu & Zhu (2018) proposed Reinforced Continual Learning (RCL) approach that searches for the best neural architecture for each coming task via reinforcement learning strategies. Like other expandable networks, this approach still suffers from higher model complexity than non-expandable networks. Also reducing its training time is a major concern.

Another category maintain a fixed network architecture with large capacity through learning. Oquab et al. (2014) proposed a weight transfer method by freezing early layers while cloning and fine-tuning later layers on the new task. In "synaptic" memory methods (Kirkpatrick et al., 2017; Zenke et al., 2017), learning rates are adjusted to minimize updates in weights important for previous tasks. Lee et al. (2017) proposed to incrementally match the moment of the posterior distribution of the neural network trained on the first and the second task, respectively. In Zeng et al. (2018) an Orthogonal Weights Modification (OWM) was proposed to ensure that new learning processes will not interfere with previously learned tasks, but can not enable backward knowledge transfer. In Shin et al. (2017); Kemker & Kanan (2018) previously acquired knowledge can be acquired by replaying of pseudo-data generated by Generative Adversarial Networks (GANs) or generative autoencoders, without revisiting actual past data for the coming task. Several "episodic" memory approaches (Jung et al., 2016; Li & Hoiem, 2016; Triki et al., 2017; Rebuffi et al., 2017) aim at storing and replaying examples from past tasks to keep predictions invariant via distillation (Hinton et al., 2015). GEM (Lopez-Paz & Ranzato, 2017) belongs to this class of episodic memory approaches, but unlike prior work, allows for backward transfer.

Comparing to these recent approaches on continual learning, GEM is relatively lightweight, computationally efficient and capable of achieving high accuracies. However, there is still room to improve GEM with respect to the limitations stated in Section 1.

## 3 A Brief Review of GEM

Most of the literature on continual learning (Rusu et al., 2016; Fernando et al., 2017; Kirkpatrick et al., 2017; Rebuffi et al., 2017) focused on a traditional setting: (1) the number of tasks is small; (2) the number of examples per task is large; (3) the learner allows for several passes over the examples per task; (4) the average performance across all tasks serves as the only metric. In contrast, a "more human-like" setting was considered in GEM: (1) the number of tasks is large; (2) the number of training examples is small for each task; (3) the learner observes the examples of per task only once; (4) knowledge transfer in backward and forward directions is assessed. Therefore following this new setting, the learner will observe the continuum of data in a manner of example by example

$$(x_1, t_1, y_1), \ldots, (x_i, t_i, y_i), \ldots, (x_n, t_n, y_n),$$

where each triplet $(x_i, t_i, y_i)$ contains a input vector, a task index, and a target vector. Note that any order on the tasks can be allowed.

The optimizing problem of GEM (Lopez-Paz & Ranzato, 2017) is

$$\min_{w} \frac{1}{2} \|g - w\|_2^2$$
$$s.t. \ \langle w, g_k \rangle \geq 0, \ \forall k < t,$$
(1)

where $g$ is the gradient vector calculated on the batch data of the current task, and $g_k$ here is the gradient on the episodic memory $M_k$ of the past task $k$. Note that the episodic memory $M_k$ stores a subset of the observed examples from task $k$. If the gradient update $g$ of the current batch fulfills all the constraints, the learner directly uses it to update like the usual gradient descent method. Otherwise, we should find a projection of $g$ in $L_2$-norm distance to satisfy any constraint related each previous task, and then use the solution $w$ to update the network weights.

We obtain a standard form of Quadratic Program (QP) by expanding the squared objective and discarding the constant term $g^T g$:

$$\min_{w} \frac{1}{2} w^T w \ - \ g^T w$$
$$s.t. \ Gw \geq 0,$$
(2)

where $G = (g_1, \ldots, g_{t-1})$. The primal QP problem can be translated into its dual problem (Dorn, 1960), which is the final optimizing problem:

$$\min_{\alpha} \frac{1}{2} \alpha^T G G^T \alpha \ + \ (Gg)^T \alpha$$
$$s.t. \ \alpha \geq 0,$$
(3)

We can recover the projected gradient update as $w^* = G^T \alpha^* + g$ where $\alpha^*$ is the solution to the dual problem. Note that solving the dual problem is computationally efficient since there are only $t - 1$ variables in $\alpha$, much smaller than the number of weights in a neural network.

## 4 Our Improvements

We propose three improvements to GEM: a technique to choose examples which can represent the past tasks better — **support examples**; a technique to tolerant some slight violation in gradient constraints on past tasks — **soft gradient constraints**; a technique to aggressively encourage gradient descend over previous tasks — **positive backward transfer**.

### 4.1 Support Examples

The first improvement is inspired from the idea of margins in Support Vector Machines (SVMs) (Williams, 2003). Due to the KKT theorem, a small set of support vectors obtained from a Quadratic

Program determines the decision boundary of SVM for binary classification. In the context of continual learning, the decision is crucial to identify what samples should be chosen in the episodic memory, since we can only store a limited budget of representative samples from previous task. In SVMs support vectors arise naturally as a solution from the QP problem of SVM; however such optimization problem does not apply to neural networks. In GEM a simple method is adopted, that is, the last $m$ examples of task $t$ are stored into the episodic memory $M_t$. Also the examples are shuffled randomly, which means that the storage is essentially random and ad hoc.

In this work, we propose an idea of choosing samples based on the notion of margins. We assume that the problem to solve here is multi-class classification, so the prediction from the network's forward propagation can be available. Suppose the output prediction $h(x, t, y)$ on a triplet example $(x, t, y)$ is computed through a softmax activation function, with value falling in the range $[0, 1]$. The following gives the definition of margins:

$$\text{margin} := h(x, t, y) - \max_{y' \neq y} h(x, t, y'). \tag{4}$$

When the classifier $h$ makes a correct prediction, the margin is positive. Otherwise when $h$ misclassifies the example, we have margin $\leq 0$. A margin value indicates the confidence of the prediction of an triplet example: larger the margin is in magnitude, more confidence we have in the prediction.

In order to alleviate catastrophic forgetting, the most important thing is to ensure that the examples of past tasks can be correctly classified to retain the prediction performance. Usually we have a much smaller budget for the episodic memory than the number of examples per task. For instance, in experiment on CIFAR100, the number of total samples is 2500 per task, while the episodic memory is 256 per task. On the one hand, we can not afford to assign the memory to the examples whose margins are smaller than 0, which means they could not be classified correctly in the past task or even these examples are outliers. On the other hand, for the examples whose margins are relatively large, indicating that the network can classify them with a strong confidence, these example should not be put into the memory either. We define the *Support Examples* as those examples being classified correctly but with a limited confidence. In experiments we simply use a confidence interval to select the support examples (we specify the interval as hyperparameters per task). In this way, the episodic memory can be used more efficiently, and at the same time, we argue that the chosen support examples can represent the past tasks better.

## 4.2 SOFT GRADIENT CONSTRAINTS

This idea is inspired by the soft-margin technique in SVMs, which is a compromise to some wrongly classified examples by linear classifiers. In continual learning, the difficulty of optimizing the gradient to be consistent to every gradient direction of past tasks is increasing along the process of learning new tasks. The constraints in (2) mean that we want the optimized gradient must have a similar direction with all of the gradients of past tasks, which is too restrictive, especially when the number of tasks is large or the tasks are competing or conflicting.

So we propose the following soft gradient constraints:

$$\begin{aligned} \min_{w} \quad & \frac{1}{2}w^T w \; - \; g^T w \; + C\Sigma_{k=1}^{t-1}\xi_k, \\ s.t. \; & Gw \geq -\xi, \;\; \xi_k \geq 0 \;\; \forall k < t, \end{aligned} \tag{5}$$

$\xi = (\xi_1, \ldots, \xi_{t-1})$ is the slack vector, $C$ is the trade-off parameter (the bigger, the stricter for $w$ to satisfy the constraints). Once any original constraint in $Gw \geq 0$ is violated, we impose a penalty in the objective. With the KKT theorem, we have the following dual problem as $\alpha$ is the Lagrange multiplier vector:

$$\begin{aligned} \min_{\alpha} \quad & \frac{1}{2}\alpha^T GG^T \alpha \; + \; (Gg)^T \alpha, \\ s.t. \; & 0 \leq \alpha \leq C1_n. \end{aligned} \tag{6}$$

Like in SVMs, the hyperparameter $C$ needs to be tuned. For the first several tasks, it is relatively easy to make the gradient update completely satisfy the constraints on all the past tasks. When the number of past tasks is increasing, it is much more difficult to have the similar direction to all the past tasks' gradient simultaneously. So we need to loose the constraint to some extent.

Here, we propose a method just like the learning rate decay in neural network training: we can initialize $C$ to a relatively big value, which should ensure a very strict constraint for first several tasks, while let $C$ decay along the process of training. In other words, after finishing training a certain number of tasks, we set $C$ smaller to tolerant more freedoms in constraint violations. This trick will maintain a good performance, and at the same time make the model more robust.

The authors of GEM claimed that adding a small offset $\gamma \geq 0$ to the Lagrangian multipliers in (2) can enable the network to perform better. However, this is an ad-hoc practice and lacks rationality: it is not faithfully follow their mathematical formulation. In our method, we can just avoid this ad-hoc practice and guarantee that just use the original constraints with slack variables will produce a good performance. Our soft constraints offer a better explanation to this issue.

### 4.2.1 POSITIVE BACKWARD TRANSFER

GEM assumes that if the inner product of the proposed gradient update and the gradients of past tasks is no less than zero, the current update will not cause catastrophic forgetting. Some extreme scenarios might occur that the found gradient projection is orthogonal to all the gradients of previous tasks, indicating that there is no contribution to backward transfer. Further, the non-negative constraints in inner products only require the angles between two gradient vectors lying in the interval $[-90°, 90°]$, totally ignoring the cosine similarity in magnitude.

In our opinion, we should specify the magnitude of the inner products of two gradient vectors for enhancing positive backward transfer to previously solved tasks. We can modify (2) to impose magnitude penalties:

$$\min_{w} \frac{1}{2}\|g - w\|_2^2$$
$$s.t. \ \langle w, g_k \rangle \geq \varepsilon \|w\| \|g_k\|, \ \forall k < t, \ \varepsilon \geq 0.$$

This new constraint imposes cosine similarity actually, so it can represent the angle and the magnitude simultaneously. Because we do not know $\|w\|$ in advance, we can use $\|g\|$ as a proxy since $w$ is the projection of $g$:

$$\min_{w} \frac{1}{2}\|g - w\|_2^2 \tag{7}$$
$$s.t. \ \langle w, g_k \rangle \geq \varepsilon \|g\| \|g_k\|, \ \forall k < t, \ \varepsilon \geq 0.$$

Similarly we get the final dual optimizing problem, which includes soft gradient constraints.

$$\min_{\alpha} \frac{1}{2}\alpha^T G G^T \alpha \ + \ (Gg \ + \ \varepsilon \|g\| \|g_k\| 1_n)^T \alpha, \tag{8}$$
$$s.t. \ 0 \leq \alpha \leq C 1_n.$$

After we solve $\alpha$, which has the variables equal to the number of past tasks, we can use $w = G^T \alpha + g$ to get the modified gradient and use it to update current network.

Here $\varepsilon$ is a hyperparameter. Similarly, the first several tasks are easier for network to enable positive backward transfer. And when the number of tasks increase, we can reuse the idea of 'decay' to weaken the magnitude of inner-product similarity, approaching zero gradually.

## 5 EXPERIMENTS

### 5.1 DATASETS AND SETTINGS

We do experiments on two dataset same to GEM (Lopez-Paz & Ranzato, 2017): *MNIST Permutations* (Kirkpatrick et al., 2017) and *Incremental CIFAR100* (Rebuffi et al., 2017). Besides, we also do experiment on *MNIST Mix* (Xu & Zhu, 2018),

1. The *MNIST Permutations* dataset is derived from the famous dataset on handwritten digits (Le-Cun et al., 1998), which randomly generates different pattern of pixel permutation for each task to exchange the position of the original images of MNIST.

2. The *Incremental CIFAR100* dataset is derived from CIFAR100 (Krizhevsky, 2009), which averagely devides the whole classes in CIFAR100 into $T$ parts, and $T$ here is the number of tasks. So in each task, there is $100/T$ different classes that are not included in other tasks.

3. The *MNIST Mix* dataset is also derived from MNIST, which has 10 *MNIST Permutations* ($P1, P2, ..., P10$) and 10 *MNIST Rotations* ($R1, R2, ..., R10$) (Lopez-Paz & Ranzato, 2017), whose image is rotated by a fixed angle from original image of MNIST. The consecutive tasks are arranged in the order $P1, R1, P2, R2, ..., P10, R10$.

To compare our method to the original GEM, we use the same settings on datasets. The number of tasks is set as $T = 20$. On the MNIST Permutations dataset there are 1000 examples per task and On the Incremental CIFAR100 dataset there are 2500 examples per task. In the basic comparison, the size of the total budget on episodic memory is set as 5120, which means that we can store 256 examples for each task.

The network architectures are also same to the experiments of the original GEM: for MNIST Permutations, we use fully-connected neural networks with two hidden layers of 100 ReLU units; for Incremental CIFAR100, we use a smaller version of ResNet18 (He et al., 2016), which has three times less feature maps for each layer than the orginal ResNet18 and has a final linear classifier for each task.

## 5.2 COMPETING METHODS

To solve the optimizing problem of soft gradient constraint, we have to change the toolbox in original code of GEM as cvxpy. So we conduct experiments both on our run of the original GEM and the GEM with the cvxpy toolbox.

Besides, we compare with other several methods:

1. *Single*, a single predictor for all the tasks.

2. *Multimodal*, a multimodal predictor, whose architecture is same to "single" and has a dedicated input layer for each task (only tested on MNIST Permutations dataset).

3. *Independent*, a network that there is a specific independent predictor for each task which has the same architecture as "single" while the number of hidden units are T times less than "single".

4. Elastic Weight Consolidation (EWC) (Kirkpatrick et al., 2017).

5. iCARL (Rebuffi et al., 2017), the class-incremental learner uses episodic memory and a nearest-exemplar algorithm (tested only on Incremental CIFAR100).

These compared methods are same to the settings in (Lopez-Paz & Ranzato, 2017). In Basic Comparison, we report the results reported in original paper of GEM, as well as our run of the original GEM, the GEM using cvxpy toolbox and our improved method. In other experiments, the method GEM is always the version of using cvxpy.

## 5.3 EVALUATION METRICS

The evaluations we use are average accuracy and positive backward transfer. The backward transfer can be calculated as:

$$\text{Backward Transfer} := \frac{1}{T-1} \sum_{i=1}^{T-1} (A_{Ti} - A_{ii})$$

where $A_{ij}$ is the accuracy of task $j$ after learning of task $i$.

## 5.4 BASIC COMPARISON

Table 1 shows the basic comparison results. To be succinct, MNIST Permutation is denoted by MNIST_P and MNIST Mix is denoted by MNIST_M. We can see that our method can perform better both on average accuracy and positive backward transfer than the original GEM on all three datasets. For example on CIFAR100, our method improves from 66.48% (our run of GEM) to

Table 1: Results in basic comparison.

| | Metrics | single | independent | multimodal | EWC |
|---|---|---|---|---|---|
| MNIST_P | Average Accuracy | 0.5742 | 0.3700 | 0.7158 | **0.6185** |
| | Backward Transfer | -0.2220 | 0.0000 | -0.0683 | **-0.1653** |
| | Metrics | GEM | GEM (rerun) | GEM-cvx | Ours |
| MNIST_P | Average Accuracy | 0.8260 | 0.8232 | 0.8213 | **0.8331** |
| | Backward Transfer | 0.0247 | 0.0386 | 0.0230 | **0.0409** |
| | Metrics | single | independent | iCARL | EWC |
| CIFAR100 | Average Accuracy | 0.4666 | 0.4415 | 0.5359 | **0.4984** |
| | Backward Transfer | -0.1240 | 0.0000 | -0.0405 | **-0.0799** |
| | Metrics | GEM | GEM (rerun) | GEM-cvx | Ours |
| CIFAR100 | Average Accuracy | 0.6783 | 0.6648 | 0.6703 | **0.6876** |
| | Backward Transfer | 0.0042 | 0.0138 | 0.0147 | **0.0403** |
| | Metrics | single | independent | multimodal | EWC |
| MNIST_M | Average Accuracy | 0.7206 | 0.5275 | 0.7656 | **0.7285** |
| | Backward Transfer | -0.1107 | 0.0000 | -0.0194 | **-0.0885** |
| | Metrics | GEM (our run) | GEM-cvx | Ours | |
| MNIST_M | Average Accuracy | 0.8521 | 0.8446 | **0.8537** | |
| | Backward Transfer | 0.0246 | 0.0129 | **0.0237** | |

Table 2: Hyperparameters of basic comparison

| | Support examples | Soft gradient constraint | Positive backward transfer |
|---|---|---|---|
| MNIST_P | $[0.1, 0.8]$ | $\infty$ & $*1$ per 20 tasks | 0.2 & $*0.8$ per 3 tasks |
| CIFAR100 | $[0.2, 0.8]$ | 5 & $*0.6$ per 3 tasks | 0.2 & $*0.8$ per 3 tasks |
| MNIST_M | $[0.1, 0.8]$ | $\infty$ & $*1$ per 20 tasks | 0.25 & $*0.8$ per 5 tasks |

68.76%, with backward transfer growing from 1.38% to 4.03%. Table 2 lists the hyperparamters used in experiments.

Figure 1 reports the evolution curves of average accuracy for all tasks and the accuracy of the first task as well. The details of numerical values are given in Appendix A. The evolution of average accuracy of all tasks indicates the comprehensive performance of the method. Our method perform best comparing to all other methods on average accuracies. Also on all three datasets, our method keeps a relatively stable performance on the first task and always has improvements through the whole learning process. We think the performance evolution of the first task only provide an small aspect of the comparison; improvements of average accuracy could be more important.

## 5.5 PERFORMANCE ON DIFFERENT MEMORIES

We also do some experiments on different budget of memories for the three datasets, with results shown in Table 3. We can see that larger budgets of memories can enhance the accuracy, and our method improves the original GEM on all settings. Table 4 shows the hyperprameters for our method.

## 5.6 ABLATION STUDY

We conduct ablation study to figure out the contribution of each improvements, with results shown in Figure 2. The details of numerical results are reported in Appendix B.

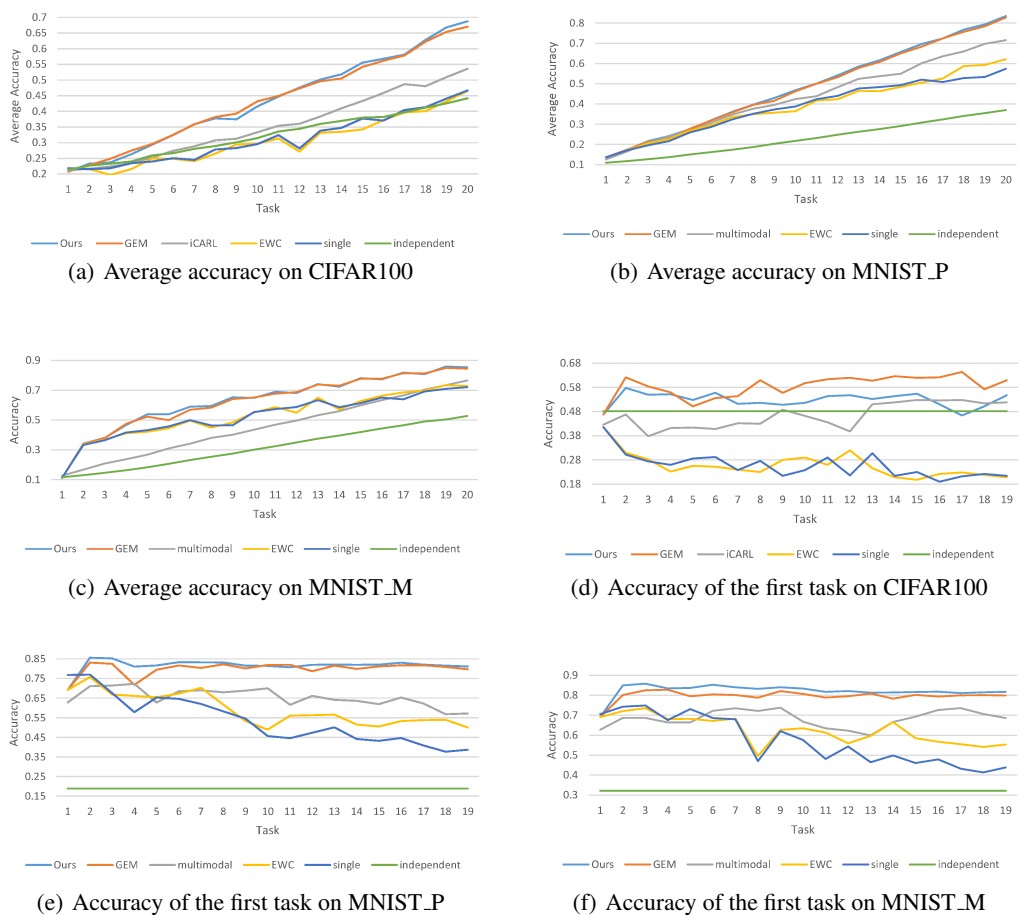

Figure 1: Evolution curves along with more tasks.

Table 3: Average accuracy in different memories

|  | Memories | GEM-cvx | Ours |
|---|---|---|---|
|  | 256 | 0.8213 | 0.8351 |
| MNIST_P | 128 | 0.8046 | 0.8096 |
|  | 64 | 0.7636 | 0.7762 |
|  | 256 | 0.6703 | 0.6876 |
| CIFAR100 | 128 | 0.6489 | 0.6556 |
|  | 64 | 0.6193 | 0.6248 |
|  | 256 | 0.8446 | 0.8537 |
| MNIST_M | 128 | 0.8165 | 0.8309 |
|  | 64 | 0.8072 | 0.8169 |

We can see that using the positive backward transfer separately can give an improvement, but the support examples and soft gradient constraint can not independently improve the performance. However, support examples can give an improvement combined with soft gradient constraint or positive backward transfer, while soft gradient constraint can probably produce a further promotion with the combination to the other two methods. So the combination of support examples and positive backward transfer can provide stable improvements, whereas the contribution of soft gradient constraint depends on the dataset.

Table 4: Hyperparameters of different memories

|         | Memories | Support examples | Soft gradient constraint | Positive backward transfer |
|---------|----------|------------------|--------------------------|----------------------------|
| MNIST_P | 256      | $[0.1, 0.8]$     | $\infty$ & $*1$ per 20 tasks | 0.2 & $*0.8$ per 3 tasks |
|         | 128      | $[0.15, 0.89]$   | $\infty$ & $*1$ per 20 tasks | 0.2 & $*0.8$ per 3 tasks |
|         | 64       | $[0.15, 0.85]$   | $\infty$ & $*1$ per 20 tasks | 0.2 & $*0.8$ per 3 tasks |
| CIFAR100 | 256     | $[0.2, 0.8]$     | 5 & $*0.6$ per 3 tasks   | 0.2 & $*0.8$ per 3 tasks |
|         | 128      | $[0.2, 0.8]$     | 5 & $*0.6$ per 3 tasks   | 0.2 & $*0.8$ per 3 tasks |
|         | 64       | $[0.2, 0.8]$     | 10 & $*0.6$ per 3 tasks  | 0.2 & $*0.8$ per 3 tasks |
| MNIST_M | 256      | $[0.1, 0.8]$     | $\infty$ & $*1$ per 20 tasks | 0.25 & $*0.8$ per 5 tasks |
|         | 128      | $[0.1, 0.8]$     | $\infty$ & $*1$ per 20 tasks | 0.25 & $*0.8$ per 5 tasks |
|         | 64       | $[0.15, 0.9]$    | $\infty$ & $*1$ per 20 tasks | 0.25 & $*0.8$ per 5 tasks |

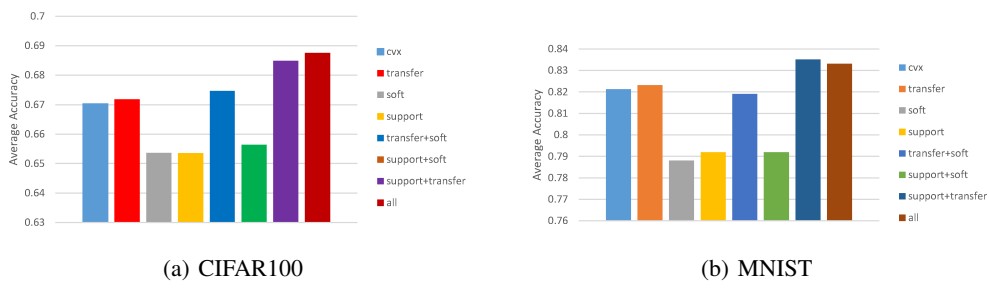

(a) CIFAR100          (b) MNIST

Figure 2: Left: Ablation study on CIFAR100. Right: Ablation study on MNIST.

## 6 CONCLUSION

We notice the fact of the precious space of memories in continual learning and the incremental difficulty through progressive learning. The GEM model is relatively lightweight, computationally fast and robust to yield high accuracy, so we propose three techniques to improve GEM. Specifically, we propose support examples to make good use of memories, formulate soft gradient constraints to accommodate competing tasks, present cosine similarity to encourage positive backward transfer over previously solved tasks. From experiments we can see the performance improvements of our techniques over the original GEM, obtaining better average accuracy and higher backward transfer. Also our method increase little computational burdens, with the cost of several hyper-parameters needing to tune. Therefore we argue that our improvements can promote GEM to be applied to more practical applications.

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

## A  A RESULTS IN ALL TASKS' AVERAGE ACCURACY AND THE ACCURACY OF THE FIRST TASK

Table 5: Results of all tasks' average accuracy in CIFAR100

| | | | | | | | | | | |
|---|---|---|---|---|---|---|---|---|---|---|
| Ours | 0.2074 | 0.2328 | 0.2368 | 0.2629 | 0.294 | 0.3248 | 0.3589 | 0.3777 | 0.3744 | 0.4158 |
| | 0.4472 | 0.4759 | 0.5014 | 0.5181 | 0.5555 | 0.568 | 0.581 | 0.6283 | 0.668 | 0.6876 |
| GEM | 0.2074 | 0.2284 | 0.2485 | 0.2748 | 0.296 | 0.325 | 0.3583 | 0.3819 | 0.3924 | 0.4318 |
| | 0.4493 | 0.4733 | 0.496 | 0.5049 | 0.5419 | 0.5617 | 0.5788 | 0.6224 | 0.6538 | 0.6703 |
| iCARL | 0.2111 | 0.2167 | 0.2236 | 0.2342 | 0.2532 | 0.274 | 0.2882 | 0.3076 | 0.3123 | 0.3333 |
| | 0.3536 | 0.3606 | 0.3835 | 0.4096 | 0.4332 | 0.4589 | 0.4867 | 0.4805 | 0.509 | 0.5359 |
| EWC | 0.2189 | 0.2164 | 0.1968 | 0.2157 | 0.2499 | 0.2486 | 0.2411 | 0.2649 | 0.2936 | 0.2964 |
| | 0.3134 | 0.2716 | 0.3318 | 0.3346 | 0.3424 | 0.3718 | 0.3968 | 0.4007 | 0.4314 | 0.4657 |
| single | 0.2189 | 0.2152 | 0.2183 | 0.2343 | 0.24 | 0.2503 | 0.2449 | 0.2782 | 0.2827 | 0.2953 |
| | 0.3243 | 0.2816 | 0.3382 | 0.3474 | 0.377 | 0.3703 | 0.4043 | 0.4134 | 0.4406 | 0.4666 |
| independent | 0.2139 | 0.2268 | 0.233 | 0.2401 | 0.2595 | 0.2666 | 0.2802 | 0.2894 | 0.3009 | 0.3147 |
| | 0.3356 | 0.344 | 0.3599 | 0.3692 | 0.38 | 0.3822 | 0.397 | 0.4119 | 0.4256 | 0.4415 |

Table 6: Results of the first task's accuracy in CIFAR100

| | | | | | | | | | | |
|---|---|---|---|---|---|---|---|---|---|---|
| Ours | 0.468 | 0.578 | 0.55 | 0.552 | 0.528 | 0.558 | 0.512 | 0.516 | 0.508 | 0.516 |
| | 0.544 | 0.548 | 0.532 | 0.544 | 0.554 | 0.51 | 0.464 | 0.502 | 0.548 | |
| GEM | 0.468 | 0.622 | 0.584 | 0.56 | 0.502 | 0.536 | 0.544 | 0.61 | 0.558 | 0.598 |
| | 0.614 | 0.62 | 0.608 | 0.626 | 0.62 | 0.622 | 0.644 | 0.572 | 0.61 | |
| iCARL | 0.426 | 0.468 | 0.378 | 0.412 | 0.414 | 0.408 | 0.432 | 0.43 | 0.488 | 0.462 |
| | 0.436 | 0.398 | 0.51 | 0.518 | 0.528 | 0.526 | 0.528 | 0.514 | 0.518 | |
| EWC | 0.416 | 0.31 | 0.282 | 0.232 | 0.256 | 0.252 | 0.24 | 0.23 | 0.28 | 0.29 |
| | 0.26 | 0.32 | 0.246 | 0.208 | 0.198 | 0.222 | 0.228 | 0.218 | 0.208 | |
| single | 0.416 | 0.302 | 0.274 | 0.26 | 0.286 | 0.292 | 0.238 | 0.276 | 0.214 | 0.238 |
| | 0.29 | 0.216 | 0.308 | 0.214 | 0.23 | 0.19 | 0.212 | 0.222 | 0.214 | |
| independent | 0.482 | 0.482 | 0.482 | 0.482 | 0.482 | 0.482 | 0.482 | 0.482 | 0.482 | 0.482 |
| | 0.482 | 0.482 | 0.482 | 0.482 | 0.482 | 0.482 | 0.482 | 0.482 | 0.482 | |

Table 7: Results of all tasks' average accuracy in MNIST Permutations

| | | | | | | | | | | |
|---|---|---|---|---|---|---|---|---|---|---|
| Ours | 0.1364 | 0.1733 | 0.2165 | 0.2419 | 0.2754 | 0.3197 | 0.3612 | 0.3971 | 0.4302 | 0.468 |
| | 0.5052 | 0.5396 | 0.5848 | 0.6189 | 0.6557 | 0.6948 | 0.7226 | 0.7645 | 0.7856 | 0.8331 |
| GEM | 0.1364 | 0.1741 | 0.2147 | 0.2315 | 0.2772 | 0.3187 | 0.3579 | 0.3956 | 0.4157 | 0.4625 |
| | 0.5 | 0.5323 | 0.5783 | 0.6082 | 0.6498 | 0.6837 | 0.7243 | 0.7567 | 0.7844 | 0.8276 |
| multimodal | 0.1256 | 0.1648 | 0.21 | 0.2394 | 0.2667 | 0.3081 | 0.3471 | 0.3765 | 0.3958 | 0.4242 |
| | 0.4382 | 0.4831 | 0.5239 | 0.5373 | 0.5491 | 0.6017 | 0.6358 | 0.6605 | 0.6974 | 0.7158 |
| EWC | 0.1364 | 0.1704 | 0.205 | 0.2271 | 0.2689 | 0.2954 | 0.3355 | 0.3495 | 0.3576 | 0.3646 |
| | 0.4177 | 0.4234 | 0.4646 | 0.4633 | 0.4843 | 0.5057 | 0.526 | 0.5874 | 0.5932 | 0.6213 |
| single | 0.1364 | 0.171 | 0.1959 | 0.2164 | 0.2599 | 0.2861 | 0.3235 | 0.3518 | 0.3732 | 0.3878 |
| | 0.4238 | 0.4399 | 0.4767 | 0.4838 | 0.493 | 0.52 | 0.5089 | 0.5283 | 0.534 | 0.5742 |
| independent | 0.1094 | 0.1176 | 0.1267 | 0.1374 | 0.1502 | 0.1618 | 0.1735 | 0.1873 | 0.2037 | 0.2178 |
| | 0.2317 | 0.2478 | 0.2631 | 0.276 | 0.2912 | 0.3083 | 0.3236 | 0.341 | 0.3547 | 0.37 |

Table 8: Results of the first task's accuracy in MNIST Permutations

| Method | | | | | | | | | | |
|---|---|---|---|---|---|---|---|---|---|---|
| Ours | 0.691 | 0.8559 | 0.8527 | 0.8109 | 0.8167 | 0.8331 | 0.8321 | 0.8314 | 0.8159 | 0.82 |
| | 0.812 | 0.8036 | 0.8154 | 0.8183 | 0.8179 | 0.825 | 0.8161 | 0.8194 | 0.8178 | |
| GEM | 0.691 | 0.831 | 0.8246 | 0.7178 | 0.7946 | 0.8164 | 0.8035 | 0.8226 | 0.8016 | 0.8188 |
| | 0.8194 | 0.7867 | 0.8159 | 0.7991 | 0.8116 | 0.8176 | 0.8176 | 0.8086 | 0.797 | |
| multimodal | 0.6274 | 0.7109 | 0.7138 | 0.7227 | 0.6278 | 0.6846 | 0.6899 | 0.6795 | 0.688 | 0.7 |
| | 0.6155 | 0.6612 | 0.6409 | 0.6361 | 0.6192 | 0.6525 | 0.6228 | 0.5679 | 0.5712 | |
| EWC | 0.691 | 0.7576 | 0.6677 | 0.6616 | 6544 | 0.673 | 0.7021 | 0.6164 | 0.5332 | 0.4895 |
| | 0.5609 | 0.5634 | 0.5655 | 0.5151 | 0.5049 | 0.5332 | 0.5381 | 0.5394 | 0.5001 | |
| single | 0.7677 | 0.7699 | 0.6753 | 0.5782 | 0.6531 | 0.6459 | 0.6201 | 0.5827 | 0.5457 | 0.4566 |
| | 0.4455 | 0.4732 | 0.5009 | 0.4416 | 0.4323 | 0.4465 | 0.4085 | 0.3767 | 0.3865 | |
| independent | 0.1889 | 0.1889 | 0.1889 | 0.1889 | 0.1889 | 0.1889 | 0.1889 | 0.1889 | 0.1889 | 0.1889 |
| | 0.1889 | 0.1889 | 0.1889 | 0.1889 | 0.1889 | 0.1889 | 0.1889 | 0.1889 | 0.1889 | |

Table 9: Results of all tasks' average accuracy in MNIST Mix

| Method | | | | | | | | | | |
|---|---|---|---|---|---|---|---|---|---|---|
| Ours | 0.691 | 0.8502 | 0.8584 | 0.8348 | 0.8368 | 0.8532 | 0.8401 | 0.8321 | 0.8403 | 0.8339 |
| | 0.8172 | 0.8214 | 0.8134 | 0.8145 | 0.8169 | 0.8184 | 0.8107 | 0.8147 | 0.8172 | |
| GEM | 0.691 | 0.8019 | 0.8256 | 0.828 | 0.7953 | 0.8045 | 0.8019 | 0.7881 | 0.8212 | 0.8073 |
| | 0.7886 | 0.7949 | 0.8086 | 0.783 | 0.8027 | 0.7938 | 0.8001 | 0.801 | 0.7991 | |
| multimodal | 0.6274 | 0.6871 | 0.6865 | 0.6638 | 0.6638 | 0.7225 | 0.7349 | 0.7214 | 0.7379 | 0.6676 |
| | 0.6347 | 0.6224 | 0.5992 | 0.6664 | 0.6933 | 0.7262 | 0.736 | 0.7064 | 0.6864 | |
| EWC | 0.691 | 0.7205 | 0.7356 | 0.6811 | 0.6823 | 0.6718 | 0.6831 | 0.4955 | 0.6263 | 0.6342 |
| | 0.6134 | 0.5594 | 0.5966 | 0.6658 | 0.5849 | 0.5676 | 0.5547 | 0.541 | 0.5528 | |
| single | 0.7062 | 0.7431 | 0.749 | 0.6757 | 0.7315 | 0.6863 | 0.6796 | 0.4702 | 0.6201 | 0.5759 |
| | 0.4812 | 0.5436 | 0.4643 | 0.4981 | 0.4607 | 0.4785 | 0.4318 | 0.4134 | 0.4387 | |
| independent | 0.3213 | 0.3213 | 0.3213 | 0.3213 | 0.3213 | 0.3213 | 0.3213 | 0.3213 | 0.3213 | 0.3213 |
| | 0.3213 | 0.3213 | 0.3213 | 0.3213 | 0.3213 | 0.3213 | 0.3213 | 0.3213 | 0.3213 | |

Table 10: Results of the first task's accuracy in MNIST Mix

| Method | | | | | | | | | | |
|---|---|---|---|---|---|---|---|---|---|---|
| Ours | 0.1116 | 0.342 | 0.3809 | 0.4658 | 0.5388 | 0.5392 | 0.589 | 0.5948 | 0.6523 | 0.6481 |
| | 0.6897 | 0.6811 | 0.7415 | 0.7238 | 0.781 | 0.7725 | 0.8185 | 0.8082 | 0.8593 | 0.8537 |
| GEM | 0.1116 | 0.3403 | 0.3789 | 0.4749 | 0.5228 | 0.4995 | 0.5695 | 0.5829 | 0.6407 | 0.6513 |
| | 0.6765 | 0.6873 | 0.738 | 0.7303 | 0.7771 | 0.7771 | 0.8145 | 0.8126 | 0.8493 | 0.8446 |
| multimodal | 0.1272 | 0.1673 | 0.2084 | 0.2371 | 0.2672 | 0.3094 | 0.3419 | 0.3802 | 0.4011 | 0.4351 |
| | 0.4688 | 0.4966 | 0.5328 | 0.5598 | 0.5992 | 0.6338 | 0.6651 | 0.7043 | 0.7336 | 0.7656 |
| EWC | 0.1116 | 0.3362 | 0.3688 | 0.4116 | 0.4221 | 0.4449 | 0.4966 | 0.4494 | 0.4835 | 0.5506 |
| | 0.5877 | 0.5494 | 0.6503 | 0.5714 | 0.6278 | 0.663 | 0.685 | 0.6992 | 0.7335 | 0.7285 |
| single | 0.1124 | 0.3329 | 0.3649 | 0.4168 | 0.4317 | 0.4572 | 0.5002 | 0.4642 | 0.4651 | 0.5531 |
| | 0.5736 | 0.5868 | 0.6336 | 0.5857 | 0.6136 | 0.6509 | 0.6388 | 0.6912 | 0.7098 | 0.7206 |
| independent | 0.1166 | 0.1302 | 0.1465 | 0.1631 | 0.1829 | 0.2068 | 0.2322 | 0.2534 | 0.2752 | 0.3011 |
| | 0.3246 | 0.3498 | 0.3757 | 0.3967 | 0.4191 | 0.4443 | 0.465 | 0.4904 | 0.5041 | 0.5275 |

## B    B ACCURACY IN ABLATION

Table 11: Average accuracy on MNIST Permutations

| Metrics | GEM-cvx | transfer | soft | support | transfer & soft |
|---|---|---|---|---|---|
| Accuracy | 0.6703 | 0.6718 | 0.6536 | 0.6535 | 0.6747 |
| | support & soft | support & transfer | all | | |
| | 0.6564 | 0.6849 | 0.6876 | | |

