# OpenReview forum: "Revisiting Gradient Episodic Memory for Continual Learning"
_ICLR.cc/2020/Conference — Reject_

### Official Review · AnonReviewer2 · 2019-10-21
**Official Blind Review #2**

**Rating:** 1

**Review:**

This paper proposes an extension to gradient episodic memory (GEM) to improve its performance and backwards transfer. Specifically, the proposed method selects "support examples" to represent each task (versus the last M examples for GEM); introduces slack variables to ensure the constraints imposed by GEM are not too restrictive; and uses cosine similarity between sample gradients to encourage backwards transfer.

The ideas proposed are interesting, and the paper is easy to follow.

However, I have a number of concerns which I think, unfortunately, preclude publication at this point.

Primarily, while I believe the ideas are intuitive and novel, the experimental evaluation does not appear to support the claim that the proposed method significantly improves GEM (only a minor improvement is shown at best). This is further compounded by the fact that variances over multiple seeds/runs are not reported, which makes it difficult to gauge any statistically significant performance improvement.

Some claims in the paper also need to be tempered:
- The abstract suggests performance improves "remarkably", but experiments do not support this.
- The last paragraph of Section 4.2 declares that adding a offset to the Lagrangian, as suggested by the GEM authors, "is an ad-hoc practice and lacks rationality: it is not faithfully follow their mathematical formulation". This may come across as confrontational, and I don't feel the point is valid in this case, given that the soft gradient constraints introduce a similar trade-off parameter.

The language is also imprecise and conversational at times; I would suggest another read-through and careful rewording for clarity. For example, in 4.2.2, "this new constraint imposes cosine similarity actually...".

Lastly, I'm not sure about this, but does the link to the GitHub repository break anonymity? Given the language difference, I'm not sure what the url refers to; but wonder if it could be a name.

The ideas have potential, and I suggest the authors explore avenues to further improve the efficacy of the method and expand the experimentation, and improve some of the writing in terms of language and claims made.

(A note on the score: I don't think this is a 1 under the old ICLR system, but unfortunately have to recommend it be rejected in its current state)

**Experience Assessment:**

I have published one or two papers in this area.

**Review Assessment: Checking Correctness Of Derivations And Theory:**

I carefully checked the derivations and theory.

**Review Assessment: Checking Correctness Of Experiments:**

I carefully checked the experiments.

**Review Assessment: Thoroughness In Paper Reading:**

I read the paper thoroughly.

---

### Official Review · AnonReviewer1 · 2019-10-22
**Official Blind Review #1**

**Rating:** 3

**Review:**

Paper proposes three improvements upon the gradient episodic memory (GEM) method [Lopez-Paz,2017]. The three improvements address 1. A way of selecting which exemplars to store (originally random), 2. The strictness of the constraints is loosened by considering slack vectors, 3. The update is improved by promoting positive backward transfer, not only limiting the gradient to the constraints imposed by exemplars of previous tasks, but aiming to improve also for previous tasks by preferring gradients which have high cosine similarity with the gradients on exemplars. Results on CIFAR 100 and MNIST permuted are presented. Noteworthy is the improved backward transfer on CIFAR 100 when compared with GEM.

Conclusion: The proposed improvements seem sensible. I liked especially the one which aims at improved backward transfer. However, the paper should have built upon the more recent A-GEM paper. Also, the proposals show strange behavior in the ablation study, and I am not convinced they all contribute to better performance. Finally, the gain with respect to GEM is very small. I, therefore, recommend a weak reject.

1.       The authors somehow missed the more recent paper ‘EFFICIENT LIFELONG LEARNING WITH A-GEM’. Their analysis should compare with this paper. This new paper does not address the points addressed in this paper, but even so, since it obtains generally slightly better results but is much more efficient, it is a better starting point. The relaxing of the constraints (with slacking vectors) might be less efficient in A-GEM which already relaxes the constraints.
2.       Improved exemplar sampling: The authors say ‘we cannot afford to assign memory to the examples whose margins are negative’, why not ?  I would like to see this ablated.
3.       Why does the average accuracy start so low ? Are they also averaging over the unseen classes? Does it not make more sense to just average over the seen classes (those considered until the current task)?
4.       The gain with respect to GEM is very small in Figure 1.
5.       Could the authors explain their validation protocol (A-GEM also makes a point of that).
6.       I think there is a typo in the equation, should have a j index on the left side, and should be A_jn-A_nn
7.        Ablation study should be better written (it takes a while to understand to which of the three proposals the abbreviations link). The results of the ablation do not convincingly show the merits of two of the three proposals. These only work when combined with ‘threshold’. No reason/explanation is provided for this strange behavior. (error in the legend the support+soft should be green).

**Experience Assessment:**

I have published one or two papers in this area.

**Review Assessment: Checking Correctness Of Derivations And Theory:**

I assessed the sensibility of the derivations and theory.

**Review Assessment: Checking Correctness Of Experiments:**

I assessed the sensibility of the experiments.

**Review Assessment: Thoroughness In Paper Reading:**

I read the paper at least twice and used my best judgement in assessing the paper.

---

### Official Review · AnonReviewer3 · 2019-10-22
**Official Blind Review #3**

**Rating:** 1

**Review:**

This paper presents three improvements to the previous GEM algorithm: choosing support examples better (the GEM paper used a random set), incorporating soft gradient constraints, and specifying the magnitude of the dot product in the gradient optimisation problem (in order to increase positive backward transfer). They then compare the original GEM algorithm (and other baselines) with their algorithm on two datasets, consider the case where there is less memory available, and provide ablation studies for their three improvements.

I recommend to reject this paper. I am struggling to see the improvements in the results, as each algorithm is only run once on each dataset, and all the numbers seem very close together (GEM vs new algorithm). There are also a significant number of hyperparameters added by this paper: how are these tuned? I also do not understand the reasoning behind the third idea ('positive backward transfer'). I do, however, like the other two improvements and see the reasoning behind them, however I still have some misgivings. I will now elaborate on these points.

Firstly, there are no standard deviations provided for the experiments. I believe this is especially important to do for the original GEM algorithm and the proposed algorithm because the two provide extremely similar metric values. Comparing values from the original GEM paper and this paper's run of GEM indicates to me that there is a significant chance that any improvements could be within error bars. For example, on CIFAR100, the authors' run of GEM gives 66.48%, the original GEM paper reports 67.83%, and the authors' method is 68.76%.

Secondly, this paper introduces a significant number of hyperparameters over the original GEM algorithm. How are these tuned? Is there a validation set? Does having to tune these hyperparameters slow down computation significantly (how much exactly)? It would be nice to see how much more computationally expensive the new method is compared with GEM: the authors claim "little computational burdens".

I like the soft gradient constraint idea introduced in this paper. It is more principled than the hack that the original GEM paper used. The method of choosing the support set also makes sense to me, however there are many hyperparameters in this idea. Additionally, the results (Table 3) for different memory sizes are confusing. It seems like smaller memory sizes lead to less improvement over random memory. Surely cleverly choosing memory should be more beneficial when constrained to smaller memory? This indicates to me that improvements need to still be made to the idea. I also do not follow the explanation given in Section 4.2.1 for the positive backward transfer improvement. Why should the magnitude of the inner products be specified? What do you mean by "cosine similarity in magnitude"?

As a final comment, I will say that there are many typos in this paper.

**Experience Assessment:**

I have published one or two papers in this area.

**Review Assessment: Checking Correctness Of Derivations And Theory:**

I carefully checked the derivations and theory.

**Review Assessment: Checking Correctness Of Experiments:**

I carefully checked the experiments.

**Review Assessment: Thoroughness In Paper Reading:**

I read the paper thoroughly.

---

### Decision · Program_Chairs · 2019-12-19

**Decision:**

Reject

**Comment:**

This paper proposes an extension of Gradient Episodic Memory (GEM) namely support examples, soft gradient constraints, and positive backward transfer. The authors argue that experiments on MNIST and CIFAR show that the proposed method consistently improves over the original GEM.

All three reviewers are not convinced with experiments in the paper. R1 and R3 mentioned that the improvements over GEM appear to be small. R2 and R3 also have some concerns without results with multiple runs. R3 has questions about hyperparameter tuning. The authors also appears to be missing recent developments in this area (e.g., A-GEM). The authors did not provide a rebuttal to these concerns.

I agree with the reviewers and recommend rejecting this paper.